# Dormant spores sense amino acids through the B subunits of their germination receptors

Lior Artzi[1], Assaf Alon [2], Kelly P. Brock[3], Anna G. Green[3], Amy Tam[3], Fernando H. Ramírez-Guadiana [1], Debora Marks[3], Andrew Kruse [2] & David Z. Rudner [1✉]

Bacteria from the orders *Bacillales* and *Clostridiales* differentiate into stress-resistant spores that can remain dormant for years, yet rapidly germinate upon nutrient sensing. How spores monitor nutrients is poorly understood but in most cases requires putative membrane receptors. The prototypical receptor from *Bacillus subtilis* consists of three proteins (GerAA, GerAB, GerAC) required for germination in response to L-alanine. GerAB belongs to the Amino Acid-Polyamine-Organocation superfamily of transporters. Using evolutionary co-variation analysis, we provide evidence that GerAB adopts a structure similar to an L-alanine transporter from this superfamily. We show that mutations in *gerAB* predicted to disrupt the ligand-binding pocket impair germination, while mutations predicted to function in L-alanine recognition enable spores to respond to L-leucine or L-serine. Finally, substitutions of bulkier residues at these positions cause constitutive germination. These data suggest that GerAB is the L-alanine sensor and that B subunits in this broadly conserved family function in nutrient detection.

[1] Department of Microbiology, Harvard Medical School, 77 Avenue Louis Pasteur, Boston, MA 02115, USA. [2] Department of Biological Chemistry and Molecular Pharmacology, Harvard Medical School, 250 Longwood Avenue, Boston, MA 02115, USA. [3] Department of Systems Biology, Harvard Medical School, 200 Longwood Avenue, Boston, MA 02115, USA. ✉email: david_rudner@hms.harvard.edu

Most spore-forming bacteria from the orders *Bacillales* and *Clostridiales* contain several operons that encode putative 3-subunit receptors of the Ger family[1]. These receptors have been implicated in the response to a wide array of nutrients including amino acids, sugars, nucleosides, and inorganic cations. Germination does not require the active transport of these molecules into the spore and it is therefore thought that the Ger receptors function as nutrient sensors[2]. Over the last four decades Ger-type receptors have been studied in *Bacillus subtilis*[1], *Bacillus megaterium*[3,4], *Bacillus cereus*[5,6], *Bacillus anthracis*[7,8], *Bacillus licheniformis*[9,10], *Clostridium botulinum*, *Clostridium sporogenes*[11], and *Clostridium perfringens*[12,13]. In some cases, a receptor is required for spores to respond to a single germinant. In others, two Ger paralogs must both be present to respond to a mixture of nutrients[14]. In yet a third example, a single Ger receptor is required for spores to respond to two or more distinct ligands that act synergistically[15]. Although the requirement for Ger family members in spore germination has been well documented, it remains unclear whether and how these complexes monitor nutrient signals. Work in *B. subtilis* and *B. megaterium* has implicated the B subunit of these complexes in nutrient sensing[4,16,17], while mutagenesis and recent structural analysis have suggested the A subunit plays this role[18,19]. However, no study has definitively established the role of these broadly conserved complexes in germinant sensing. Here, we address these questions using the prototypical nutrient receptor GerA from *B. subtilis*.

## Results

**Structural modeling of GerAB.** The B subunit of the GerA receptor (GerAB) has low sequence identity (~21%) to the *Geobacillus kaustophilus* L-alanine transporter GkApcT, a member of the Amino Acid-Polyamine-Organocation (APC) superfamily[20]. However, the remote homology detection program HHPred[21] identified GkApcT as a top hit using GerAB as a query to search for homologs of known structure (Probability 99.95%, E-value 6.2e$^{-23}$). To investigate whether GerAB adopts a similar fold to GkApcT, we threaded GerAB onto the recently determined crystal structure of GkApcT (PDB ID: 5OQT)[20] and compared the residues with a minimum-atom distance of 5 Å in this model to evolutionary coupled (EC) residues in GerAB identified by evolutionary co-variation analysis[22,23] (Fig. 1a). Co-variation analysis relies on the fact that amino acids involved in intramolecular interactions tend to co-evolve with one another to maintain their interactions. Thus, EC residue pairs generally reside in close proximity. We analyzed >20,000 GerAB homologs and identified 330 long-range EC pairs with probability scores >0.9. We then compared these EC pairs with all residue pairs in the threaded GerAB model that are ≤5 Å apart. As can be seen in the interaction matrix in Fig. 1a, among the EC pairs (black circles), 51% were within 5 Å of each other in the threaded model (light blue circles) and 85% of the EC pairs were within 8 Å of each other (Supplementary Fig. 1a). These data strongly suggest that the overall fold is conserved between these two APC superfamily members and that the homology model is built on a correct alignment without major amino acid register errors.

GkApcT has 12 transmembrane (TM) segments and, like all APC superfamily members, two-fold inverted pseudo-symmetry[24,25]. The first and sixth TM helices in GkApcT contain characteristic discontinuities that generate the ligand-binding pocket (Supplementary Fig. 1b). Helices 3, 6, and 8 contribute amino acids that line this pocket and provide L-alanine specificity. Methionine 321 in helix 8 forms the base of the pocket and substitution to serine enabled transport of L-arginine in addition to L-alanine[20].

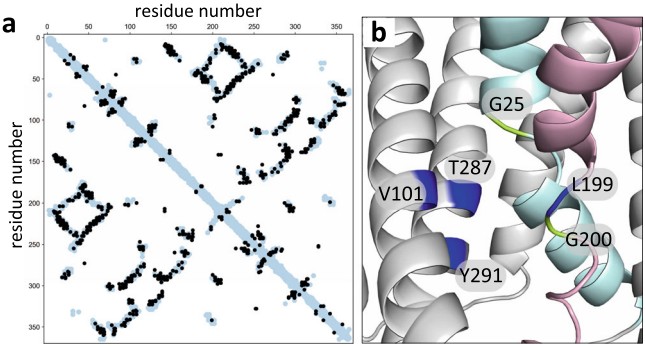

**Fig. 1 GerAB resembles the L-alanine transporter GkApcT. a** Interaction matrix comparing evolutionary coupled (EC) residue pairs in GerAB (black circles) with residue pairs that are ≤5 Å apart in the GerAB model (light blue circles) derived from the GkApcT structure. 51% of EC pairs were within 5 Å of each other in the GerAB homology model. **b** Predicted L-alanine binding pocket in GerAB. TM segments 1 (cyan) and 6 (pink) are highlighted. G25 and G200 that are predicted to generate discontinuities in these helices are shown in green. Residues predicted to line the L-alanine binding pocket are indicated in dark blue. A side-by-side comparison to the L-alanine binding pocket in GkApcT is shown in Supplementary Fig. 1b.

**Mutations in the putative ligand-binding pocket block germination.** Motivated by the structure–function analysis of GkApcT[20], we investigated the analogous residues in the predicted GerAB structure. For these studies we used a *B. subtilis* strain in which GerA was the only functional germinant receptor. Furthermore, to avoid polar effects, the three genes in the *gerA* operon were expressed separately at three ectopic chromosomal loci. Spores derived from this strain germinated in response to L-alanine with kinetics similar to wild-type (Supplementary Fig. 2 and Supplementary Table 1).

In the threaded GerAB structure, the discontinuities in TM helix 1 and 6 are generated by glycines (G25 and G200) (Fig. 1b). As a first test of the structural model, we individually substituted these residues for alanine and analyzed spore viability and germination. As can be seen in Fig. 2, both mutants had reduced spore viability and failed to germinate in the presence of L-alanine as assayed by a reduction in optical density as germinating spores transition from phase-bright to phase-dark (Fig. 2a, b). To investigate whether the mutant proteins were stably produced, we took advantage of the fact that the A and C subunits of the GerA receptor depend on GerAB for stability[16,19]. Immunoblot analysis of spores harboring GerAB(G25A) and a functional GerAC-His6 fusion indicated that the levels of both GerAA and GerAC were similar to those in spores with wild-type GerAB (Fig. 2c). By contrast, the levels of both subunits in spores harboring GerAB(G200A) were low and similar to the levels in Δ*gerAB* spores. Thus, GerAB(G25A) is produced and assembles into a stable complex with GerAA and GerAC, while GerAB(G200A) is either unstable or unable to form a complex with its partner proteins. Altogether these results are consistent with the idea that GerAB has an L-alanine-binding pocket similar to GkApcT's.

**Mutations that alter germinant specificity.** To test the hypothesis that GerAB monitors L-alanine, we sought to change germinant specificity by mutating the residues that line the putative substrate-binding site. We began by targeting tyrosine 291 in TM helix 8 that is predicted to lie at the base of the pocket, analogous to M321 in GkApcT[20]. However, spores harboring GerAB(Y291S) had reduced levels of GerAA (Supplementary Fig. 3c), indicating that the mutant protein was unstable or unable to interact with its cognate subunits. Consistent with this finding, the mutant spores failed to germinate in

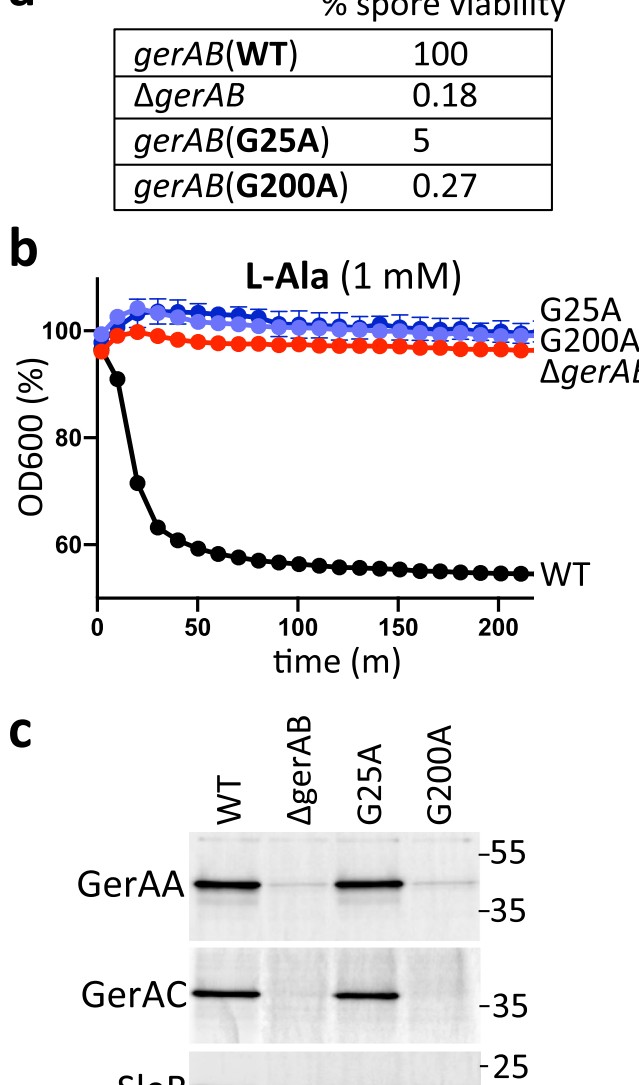

**Fig. 2 G25 and G200 are required for spore germination. a** Spore viability of the indicated strains following heat treatment (80 °C for 20 min) and colony formation on LB agar plates. **b** Spore germination in response to 1 mM L-alanine as assessed by the percent reduction in OD$_{600}$. Representative data from one of three biological replicates with two technical replicates are shown. Error bars indicate ± SD of two technical replicates. **c** Impact of GerAB variants on the stability of GerAA and GerAC in dormant spores. Immunoblot analysis of spore lysates of the indicated strains using anti-GerAA and anti-His antibodies. SleB was used to control for loading. Molecular weight markers in Kd are indicated on the right. Representative data from one of three biological replicates are shown. Source data are provided as a Source Data file.

response to L-alanine and germination and outgrowth on LB agar plates was similar to spores lacking GerAB (Supplementary Fig. 3a, b). Accordingly, we turned our attention to residues in TM helices 3, 6, and 8 that are predicted to line the ligand-binding pocket and provide specificity for L-alanine. We began by testing amino acid substitutions at V101 in helix 3 (Fig. 1b). Spores harboring GerAB(V101C) or (V101A) germinated with kinetics similar to wild-type over a range of L-alanine concentrations (Fig. 3a and Supplementary Fig. 4), indicating that these substitutions did not disrupt the putative ligand-binding pocket. We next investigated whether the

mutant spores could respond to other nutrients. Spores harboring the wild-type *gerA* locus respond to micromolar concentrations of L-alanine, low millimolar concentrations of L-valine and inefficiently to a few other amino acids when present at concentrations >25 mM[26,27]. Here, we tested a panel of 9 amino acids. Similar to wild-type, both mutants failed to respond to L-threonine, L-serine, L-asparagine, or L-arginine and weakly responded to 3 mM L-valine and 5 mM L-cysteine (Supplementary Fig. 5). However, both GerAB variants specifically and robustly responded to L-leucine (Fig. 3b and Supplementary Fig. 5). The germination assay in Fig. 3b shows the response of wild-type and mutant spores to 5 mM L-leucine. However, spores harboring GerAB(V101C) displayed partial germination in response in 0.3 mM L-leucine and full germination with 1 mM while spores with GerAB(V101A) partially responded to 1 mM L-leucine (Supplementary Fig. 6a). By contrast, spores with wild-type GerAB failed to germinate even in the presence of 50 mM L-leucine (Supplementary Fig. 6a). Importantly, immunoblot analysis revealed that GerAA and GerAC levels in the V101 mutant spores were lower than wild-type (Supplementary Fig. 7) indicating that the expanded germinant specificity was not a consequence of over-expression. Finally, we found that D-alanine, a competitive inhibitor of L-alanine[28] (Supplementary Fig. 8), also inhibited germination by L-leucine (Fig. 3d), consistent with the idea that L-leucine binds in the same pocket as L-alanine in the GerAB mutants.

Prompted by the identification of GerAB mutants with expanded germination specificity, we generated a series of mutations at L199 in TM helix 6 and T287 in TM helix 8 (Fig. 1b). Among the mutants tested, we identified one (L199S) that enabled spores to respond to L-serine (Fig. 3e). Spores harboring this mutant had a nearly full response to 3 mM L-serine and weakly responded to 1 mM (Supplementary Fig. 6c). By comparison, spores with wild-type GerAB barely germinated in the presence of 30 mM L-serine (Supplementary Fig. 6c). Importantly, the GerAB(L199S) spores responded to a panel of amino acids in a manner similar to wild-type (Supplementary Fig. 5), indicating that the change in response to L-serine was specific. Finally, we identified two mutants, GerAB(L199I) and GerAB(T287V), that like V101C and V101A were more sensitive to L-isoleucine compared to wild-type GerAB (Supplementary Fig. 5d, e and Supplementary Fig. 6b). In all cases, the levels of GerAA and GerAC in spores derived from these mutants were similar to wild-type (Supplementary Fig. 7). Collectively, these data support the idea that GerAB adopts a fold similar to GkApcT and argue that GerAB detects L-alanine in its ligand-binding pocket.

**Bulkier residues mimic L-alanine binding**. As a final test of our model, we investigated whether substituting residues that line the ligand-binding pocket with bulkier amino acids could mimic L-alanine binding and trigger premature germination in the absence of germinant signals. We generated and tested a series of amino acid substitutions at V101, L199, and T287 (Supplementary Table 2). Two such mutants, V101F and T287L, had the anticipated phenotype. Sporulating cultures of these mutants had a high percentage of phase-dark spores compared to wild-type GerAB and the ΔgerAB mutant (Fig. 4a), suggesting that the developing spores had inappropriately germinated during spore formation. Consistent with this idea, these spore cultures had reduced colony-forming units after incubation at 80 °C for 20 min (Fig. 4b) indicative of a loss in spore resistance properties. Importantly, these phenotypes were not due to over-expression of GerA complexes in the mutants (Supplementary Fig. 7). Premature germination and the production of phase-dark spores during sporulation result from Ger-dependent activation of an enzyme (SleB) that degrades the specialized peptidoglycan called

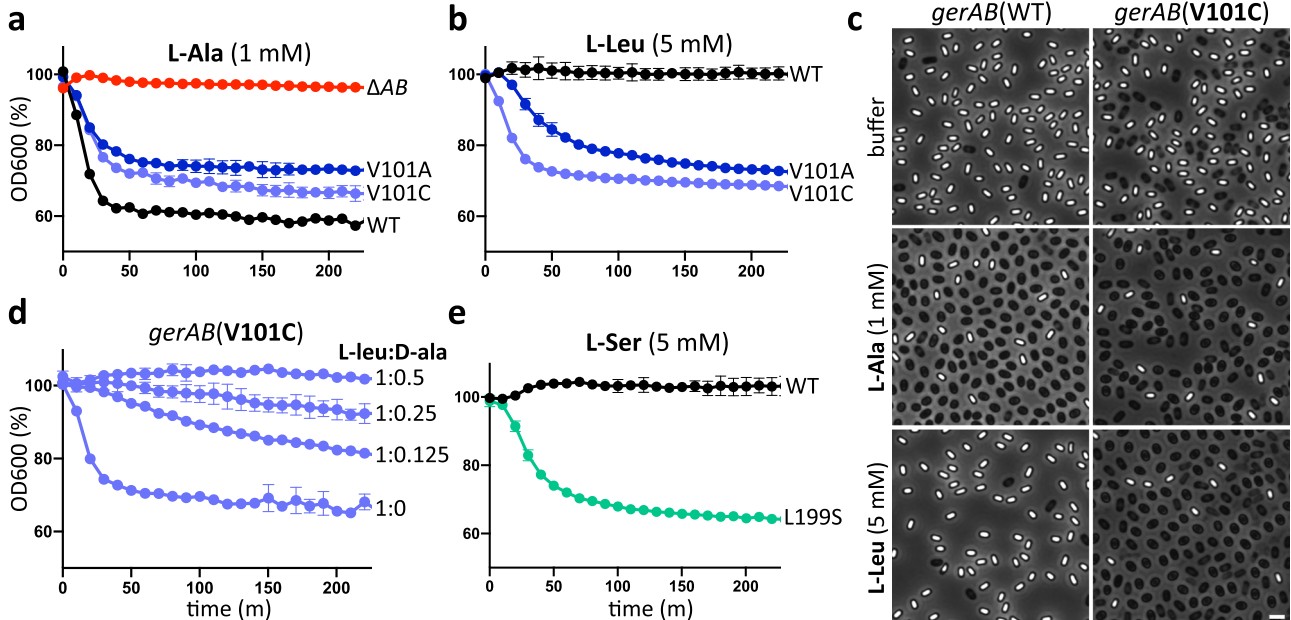

**Fig. 3 Mutations in the putative L-alanine-binding pocket of GerAB enable spores to respond to L-leucine or L-serine. a** Spores harboring *gerAB*(V101A) or *gerAB*(V101C) germinate in response to 1 mM L-alanine with kinetics similar to wild-type (WT). **b** V101A and V101C mutant spores but not wild-type also germinate in response to 5 mM L-leucine. **c** Representative phase-contrast images of wild-type and *gerAB*(V101C) spores after 100 min incubation with buffer, 1 mM L-alanine, or 5 mM L-leucine. Larger fields can be found in Supplementary Fig. 9. Scale bar, 2 μm. **d** D-alanine inhibits germination of *gerAB*(V101C) spores in response to L-leucine. Spores were incubated with 5 mM L-leucine with or without D-alanine at 0.625, 1.25, 2.5 mM, and $OD_{600}$ was monitored. The decrease in optical density in the V101 mutants was reproducibly less than wild-type in the germination assays and is likely due to spontaneous germination of a subset of the spores during their purification. **e** Spores harboring *gerAB*(L199S) germinate in response to 5 mM L-serine. Representative data from one of three biological replicates with two technical replicates are shown. Error bars in **a**, **b**, **d**, **e** indicate ± SD of two technical replicates.

the cortex that surrounds the dormant spore[29,30]. To investigate whether the phase-dark spores in the GerAB(V101F) and GerAB(T287L) cultures resulted from SleB activation, we analyzed the point mutants in a Δ*sleB* background. As can be seen in Fig. 4a, the absence of SleB largely suppressed the phase-dark-spore phenotype. Instead, many of the spores had a phase-gray appearance, which results from release of dipicolinic acid (DPA) from the spore core, an early step in GerA-dependent germination. Collectively, these data indicate that bulkier residues in the ligand-binding pocket can trigger germination providing additional evidence that GerAB functions as the L-alanine sensor and utilizes a similar ligand-binding pocket as GkApcT.

## Discussion

Altogether our results provide strong support for the model that *B. subtilis* GerA is a nutrient receptor and that the B subunit of this complex directly senses L-alanine. These findings lead us to hypothesize that the B subunits of Ger family receptors in all endospore forming bacteria function as nutrient sensors. Many of these putative receptors are required for germination in response to distinct amino acids and our data highlight how specificity could evolve through changes in the residues in and around the ligand-binding pocket. However, sugars and nucleosides are also common germinants and these nutrients similarly require Ger family members to trigger germination[1,14,31]. Interestingly, there are many sugar and nucleobase transporters in the APC superfamily including the solute sodium symporter (SSS) and nucleobase–cation–symport-1 (NCS1) families[25]. Structures of the SSS family member vSGLT[32], a sodium/galactose transporter from *Vibrio parahaemolyticus* and the NCS1 family member Mhp1, a benzyl-hydantoin transporter from *Microbacterium liquefaciens* have been solved[33–35]. the cores of these transporters

are structurally homologous to that of GkApcT with ligand-binding pockets that overlap or lie adjacent to the L-alanine binding-pocket described here (Supplementary Fig. 11). These data support the idea that B subunits in the Ger family could also sense sugars and nucleosides and raise the possibility that some germinant receptors could simultaneously bind both sugar and amino acid (or nucleoside) and thus function to integrate nutrient status in the environment. In addition, both vSGLT and Mhp1 use the inward $Na^+$ gradient to drive transport by promoting a structural rearrangement that increases substrate affinity. Some Ger receptors require $Na^+$ or $K^+$ as a co-germinant to respond to nutrients[26,36] and the $Na^+$-binding sites in vSGLT and Mhp1 could explain the mechanistic basis for this requirement. Future studies will be focused on testing whether B subunits sense sugars and nucleosides and whether these subunits can bind nutrients cooperatively.

A recent structural study of the N-terminal domain of a GerAA homolog, GerK$_3$A from *B. megaterium*, revealed that this soluble domain shares structural similarity with substrate-binding proteins from bacterial ABC transporters, albeit with a distinct order and connectivity of the structural segments[18]. This finding suggests that the GerAA subunit could also function in nutrient sensing. To detect exogenous germinants, this domain would need to face the external environment. Previous studies aimed at defining the topology of A subunits from Ger receptors in *B. subtilis* and *B. anthracis* were equivocal with one study placing the soluble N-terminal domain outside and another concluding it was intracellular and therefore inaccessible to nutrient signals[37,38]. To investigate this discrepancy, we used co-variation analysis to identify evolutionary coupled residues within GerAA and between subunits in the GerA receptor[39,40]. This analysis is consistent with a model in which the N-terminal domain of GerAA resides in the spore core (Supplementary Fig. 12a, b).

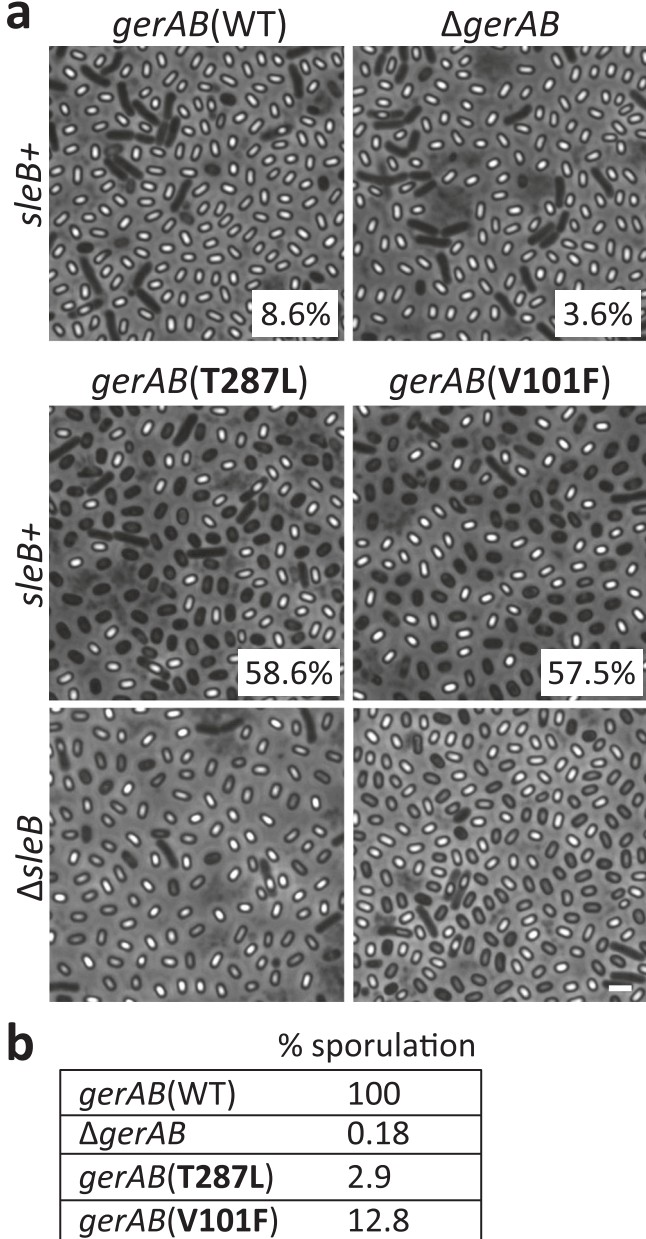

**b**

## % sporulation

| | |
|---|---|
| *gerAB*(WT) | 100 |
| Δ*gerAB* | 0.18 |
| *gerAB*(**T287L**) | 2.9 |
| *gerAB*(**V101F**) | 12.8 |

**Fig. 4 Substitutions of bulkier residues in the putative ligand-binding pocket of GerAB cause premature germination. a** Representative phase-contrast images of indicated strains after 30 h of sporulation. The majority of sporulating cells with wild-type *gerAB* or lacking *gerAB* produce phase-bright spores. By contrast, >50% of the cells harboring *gerAB* (T287L) or *gerAB* (V101F) produce phase-dark spores indicative of premature germination. Percentages of phase-dark spores are indicated in the corner of each image. Deletion of *sleB* encoding the spore cortex hydrolase that is activated during germination suppresses the phase-dark phenotype in the *gerAB* mutants. Larger fields can be found in Supplementary Fig. 10. Scale bar, 2 μm. **b** Spore viability of the strains shown in **a**. Representative data from one of three biological replicates are shown.

To more rigorously assess the topology of GerAA, we performed cysteine accessibility and protease protection assays on exponentially growing cells expressing the GerA subunits (Supplementary Fig. 12c, d). Both assays provide strong experimental support for the model that the N-terminal domain of GerAA resides in the spore core. Although we cannot rule out the possibility that the N-terminal domains of other A subunits face the

external environment, we favor a model in which the B subunits of these receptors function as the nutrient sensors with the A subunits serving as signal transducers.

Transporters by definition have to undergo a conformational change in order to function, since they cannot allow substrate to access both sides of the membrane at the same time. A receptor likewise needs to link ligand binding to a conformational change that can be detected by effector proteins. A transporter thus represents an ideal potential starting point for evolving a receptor, since it already has ligand binding and conformational change activities. The B subunits of the Ger family are not the only example of this conversion. Lysosomal amino acid levels are an important regulatory input for the mTORC1 pathway in eukaryotes, and one of the key amino acid sensors in this pathway is SLC38A9, a transporter homolog and an APC superfamily member. SLC38A9 detects lysosomal arginine[41–43], and possesses multiple activities including arginine-stimulated transport of amino acids[44] as well as acting as a guanine nucleotide exchange factor for Rag GTPases[45]. These activities have led to SLC38A9 being designated a "transceptor", combining activities typically exclusive to receptors and transporters[46]. It remains to be seen whether Ger receptor B subunits function solely as receptors, or if they also retain some transport activity.

How nutrient detection by GerAB activates germination is still unclear. We envision a scenario in which a conformational change in GerAB upon L-alanine binding modulates GerAA, which directly or indirectly triggers release of DPA from the spore core and activates the cortex lytic enzymes. GerAC is a lipoprotein with unknown function. However, the interaction between GerAC and GerAB/GerAA predicted by our co-variation analysis (Supplementary Fig. 12b) is reminiscent of the mammalian heteromeric amino acid transporters (HATs). These transporters are composed of two subunits: a ligand-binding transporter of the SLC7 family, part of the APC superfamily, and a type II membrane protein of the SLC3 family that contains a large extracellular domain. Recent structural studies of human L-type amino acid transporter 1 of the SLC7 family in complex with CD98hc of the SLC3 family[47,48] indicate that the extracellular domain of the SLC3 subunit sits on top of the SLC7 transporter and is essential for transport activity. The contacts between the two subunits suggest that the extracellular domain of CD98hc modulates the path of transported substrates. By analogy, we speculate that GerAC subunits could help channel nutrient signals to GerAB through the spore's integument layers.

It is noteworthy that *Clostridioides difficile* and other Peptostreptococcaceae family members do not encode Ger type receptors[49,50]. Instead, these endospore formers encode two soluble pseudoproteases CspC and CspA that have been implicated in germinant sensing. CspC and CspA are thought to hold a subtilisin-like protease CspB inactive within the integument layers of the dormant spore. In response to germinants, inhibition is relieved allowing CspB to cleave and activate the spore cell wall degrading enzyme SleC. Germination of *C. difficile* spores in the colon is triggered by mammalian-specific bile acids in combination with amino acids or $Ca^{2+}$ co-germinants. Recent studies have uncovered point mutations in *cspC* that allow the bile acid chenodeoxycholic acid to act as a spore germinant rather than an inhibitor of germination[51]. Mutations have also been identified in *cspA* that bypass the requirement for co-germinants[52]. These findings suggest that CspC could be the bile acid sensor and CspA a sensor of co-germinants. However, unlike the GerAB point mutants that narrowly and specifically expanded amino acid recognition, the CspC point mutants have recently been found to broadly increase sensitivity to bile acids and co-germinants[53] raising the possibility that CspC alone could sense and integrate multiple signals as proposed for GerAB or that CspC and possibly

CspA function in transducing these signals rather than directly sensing them. The identification of mutations in *cspC* (and *cspA*) like the ones described here will help address these alternative models. Understanding these distinct germination pathways has the potential to uncover common principles for nutrient detection and signal transduction during exit from dormancy.

The Ger receptors have been studied for decades but whether and how they sense nutrients has been unclear. Here, we provide evidence that the B subunits of these complexes adopt the structure of APC family transporters and possess a ligand-binding pocket involved in amino acid recognition. Importantly, agonists and antagonists that interact with the substrate-binding sites have been identified for other APC superfamily members[54] suggesting that the ligand binding pockets of B subunits can be similarly targeted. Structural studies of B family members will enable their discovery and will begin to address how ligand binding triggers exit from dormancy.

## Methods

**General methods.** All strains were derived from *Bacillus subtilis* 168[55]. Cells were sporulated by nutrient exhaustion in DS medium (DSM)[56] at 37 °C for 48 h. Sporulation efficiency was determined by comparing heat resistant (80 °C for 20 min) colony forming units (CFUs) of the mutants to wild-type. All insertion–deletion mutants were generated using the *Bacillus* knock-out collection (BKE)[57] or by isothermal assembly of PCR products and direct transformation into *Bacillus subtilis*. Site-directed mutations in *gerAB* were generated by quick change. All mutants and inserts generated by PCR were sequence-verified. Lists of strains (Supplementary Table 3), plasmids (Supplementary Table 4), and oligonucleotide primers (Supplementary Table 5) used in this study can be found in supplementary information. A detailed description of the construction of strains and plasmids can be found in supplementary methods.

**Structural modeling.** An initial alignment was generated between GerAB and GkApcT using the HHPred server[21] (https://toolkit.tuebingen.mpg.de/tools/hhpred). The resulting alignment was used to construct a homology model in MODELLER[58] using the GkApcT structure as a template (PDB ID: 5OQT).

**Evolutionary co-variation analysis.** EVcouplings software (available at https://github.com/debbiemarkslab/EVcouplings)[22,23,59] version 0.0.5 (GerAA and GerAB) and 0.1.1 (GerVB) was used on multiple sequence alignments generated for GerAA, GerAB, and GerVB. Alignments were generated using the jackhmmer software[60] with five iterations against the uniref100 dataset downloaded January 2020 (GerAA and GerAB) or July 2020 (GerVB)[61] across a range of normalized bitscores. The GerAB alignment consisted of 20,525 sequences with 95.6% coverage, at least 70% non-gap characters with fragments filtered at a threshold of 70%. The GerAA, alignment consisted of 23,876 sequences with 95.9% coverage. The GerVB alignment consisted of 21,695 sequences and 96.2% coverage. Evolutionary couplings were then calculated for these alignments using pseudolikelihood maximization to infer parameters used to calculate evolutionary couplings scores. For more information about the analysis see supplementary methods. The top-ranked evolutionary couplings scores for 330 long-range (separated by at least five amino acids in sequence) residue pairs were then displayed in the contact maps shown in Fig. 1a and Supplementary Fig. 1a.

**Spore preparation and purification.** Spores were generated on DSM agar plates. Briefly, cells were grown to an $OD_{600}$ of ~0.2 in liquid medium and spread on DSM agar plates and incubated at 37 °C for 96 h. Spores were scraped off the plate and suspended in $ddH_2O$. The suspension was washed three times and then resuspended in 350 μL 20% Histodenz (Sigma-Aldrich). The suspension was layered on top of 1 mL 50% Histodenz step-gradient and centrifuged at $20,000 \times g$ for 30 min at room temperature. Mature phase-bright spores were collected from the pellet fraction, washed four times in $ddH_2O$ and stored in 1 mL of 1× phosphate buffered saline (PBS) at 4 °C.

**Germination assay.** Purified phase-bright spores, normalized to $OD_{600}$ of 1.2 in 25 mM HEPES pH 7.4, were heat-activated at 70 °C for 30 min followed by 20 min incubation on ice. Hundred microliter of the indicated amino acid solutions were dispensed into a 96-well plate. Hundred microliter of heat-activated spores were then added and the $OD_{600}$ was monitored every 2 min for 4 h using Infinite M Plex plate reader (Tecan). The 96-well plate was maintained at 37 °C with agitation between measurements. All amino acids were dissolved in 25 mM HEPES, pH 7.4. For L-alanine:D-alanine competition, L-alanine was held at 1 mM, and D-alanine concentration was varied as indicated. For L-leucine:D-alanine competition assays, L-leucine was held at 5 mM.

**Microscopy.** Purified spores were normalized to $OD_{600}$ of 1.2, heat-activated as described above, and 250 μL of the spore suspension were transferred to glass test tubes. Spores were then mixed 1:1 with 25 mM HEPES buffer pH 7.4, 2 mM L-alanine, or 10 mM L-leucine and incubated 37 °C with aeration for 100 min. The spores were then concentrated by centrifugation, washed once in $ddH_2O$, and placed on 2% agar pads made with $ddH_2O$ and analyzed by phase-contrast microscopy. To assess premature germination, cells were sporulated in 3 mL DSM for 30 h at 37 °C, concentrated by centrifugation, washed once in $ddH_2O$, and imaged on 2% $ddH_2O$ agar pads. Phase-contrast microscopy was performed using Nikon TE2000 inverted microscope equipped with Plan Apo 100×/1.4 Oil Ph3 DM objective lens and CoolSNAP HQ2 monochrome CCD camera (Photometrics). Image analysis was preformed using MetaMorph software (Molecular Devices; version 7.7).

**Immunoblot analysis.** Cells were sporulated in 25 mL DSM for 48 h at 37 °C. The cultures were incubated at 80 °C for 20 min and spores and cell debris were pelleted, washed twice with 25 mL $ddH_2O$, and resuspended in 1 mL 1× PBS. The suspension was incubated with lysozyme (1.5 mg/mL final) at 37 °C for 1 h followed by the addition of sodium dodecylsulfate (SDS) (2% final) for 30 min. Spores were washed five times with $ddH_2O$ and resuspended in 500 μL 1× PBS, supplemented with phenylmethylsulfonyl fluoride (PMSF) (1 mM final) (Sigma-Aldrich). Spores were then lysed using FastPrep (MP Biomedicals) in 2 mL tubes containing lysis matrix B (MP Biomedicals). Immediately after lysis, 500 μL of 2× Laemmli sample buffer containing 10% β-mercaptoethanol was added. After centrifugation at $20,000 \times g$ for 20 min, the supernatant was transferred to a fresh tube, and protein concentration was determined using a noninterfering protein assay (G-Biosciences). Twelve microgram of protein was resolved by SDS-PAGE on a 17.5% acrylamide gel and transferred to an Immobilon-P membrane (Millipore). The membranes were blocked with 5% bovine serum albumin in 1XPBS with 0.5% Tween 20 (PBST) and probed with anti-GerAA[62] (1:5000), anti-SleB[63] (1:5000), anti-SpoVAD[64] (1:10,000) and anti-His (1:4000) (GenScript) antibodies, diluted in 3% BSA in PBST. Primary antibodies were detected with anti-rabbit (1:3000) and anti-mouse (1:20,000) secondary antibodies coupled to horseradish peroxidase (Bio-Rad), diluted in 3% BSA-PBST, and detected with Western Lightning ECL reagent (PerkinElmer). Although SleB and SpoVAD are in the germination pathway their production and stability in dormant spores are unaffected by the presence or absence of the Ger receptors. Accordingly, they serve as reliable controls for loading. All unprocessed immunoblots are presented in the Source Data file.

**Reporting summary.** Further information on research design is available in the Nature Research Reporting Summary linked to this article.

## Data availability

All datasets generated in this study are available at https://github.com/debbiemarkslab/GerA-suppData-2021. For EVcouplings analysis, all data for alignments were collected from publicly available datasets (Uniref100) (https://www.uniprot.org/uniref/) and ENA genome location tables deposited in Figshare https://figshare.com/articles/dataset/_/16873912, with the https://doi.org/10.6084/m9.figshare.16873912. The alignment between GerAB and GkApcT was generated using the HHPred server (https://toolkit.tuebingen.mpg.de/tools/hhpred), and a homology model from the alignment was constructed in MODELLER (https://salilab.org/modeller/) using the structure of GkApcT as a template (PDB ID: 5OQT). Source data are provided with this paper.

## Code availability

The EVcouplings software used in this study is available at https://github.com/debbiemarkslab/EVcouplings.

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

## Acknowledgements

We thank all members of the Bernhardt-Rudner super-group past and present for helpful advice, discussions, and encouragement, Jeremy Amon for feedback and discussion, Paula Montero Llopis and the HMS Microscopy Resources on the North Quad (MicRoN) core for advice on microscopy; Support for this work comes from GM086466, GM127399, DARPA HR001117S0029 (D.Z.R.), and funds from the HMS Dean's Initiative (D.M., A.K. and D.Z.R.). A.G.G. was supported by an NSF Graduate Research Fellowship DGE1144152. L.A. is a Simons Foundation fellow of the Life Sciences Research Foundation. During the revision of our manuscript, DeepMind made its open-source software for the protein structure prediction program AlphaFold2 available. The AlphaFold2-predicted structure for GerAB closely resembles the threaded GerAB model used in our study with an all-atom RMSD of 3.57 Å.

## Author contributions

L.A., D.Z.R., A.K. and D.M. designed the study. K.P.B., A.G.G. and A.T. performed the computational analysis. L.A. performed the experimental work with assistance from A.A. and F.R.G. L.A. and D.Z.R. wrote the manuscript. All authors edited and approved the final manuscript.

## Competing interests

The authors declare no competing interests.
