## [Peer Review File · Nature Communications]

Dormant spores sense amino acids through the B subunits of their germination receptorsREVIEWER COMMENTS

Reviewer #1 (Remarks to the Author):

The manuscript by Artzi et al. provides compelling evidence for a model in which the B subunit of germinant receptors serves as the direct sensor of amino acids in bacterial spores. Using evolutionary co-variation analysis, the authors discovered structural homology between GerAB subunits and the Amino acid-Polyamine-Organocation (APC) superfamily. Guided by the structure of an APC family member, GkApcT, bound to its alanine ligand, the authors target the putative ligand binding region of *Bacillus subtilis* GerAB for mutagenesis. They identify several interesting classes of mutations that impact GerAB function without affecting its levels in *B. subtilis* spores: (i) mutations that ablate germinant sensing (G25A no longer responds to alanine germinant), (ii) mutations that reduce the size of a side chain and accordingly expand germinant specificity to include specific amino acids like leucine (V101C/A), serine (L199S), or isoleucine (L199I and T287V) in addition to the primary alanine germinant, and (iii) mutations that mimic germinant binding and prematurely activate germination (V101F and T287L).

The manuscript is well written and presents intriguing possible mechanisms for germinant sensing and signal transduction informed by studies of transporters in eukaryotic organisms, so it will be of wide interest to the community. However, the title and abstract are not as circumspect in terms of acknowledging that it is theoretically possible that the mutations the authors have identified affect signal transduction rather than ligand binding. Thus, the authors should temper their language slightly.

Despite lacking biochemical evidence of GerAB directly binding to amino acid ligands, the authors' identification of mutants with expanded germinant specificity provides strong genetic support for GerAB functioning as the germinant sensor. Their analysis of the germinant sensitivity by titrating different germinants and comparing the germinant specificity of their different mutants provides convincing data in support of hypothesis. Their work stands in contrast to previously published work (2019) implicating the A subunit of Ger receptors as the direct binding partner for germinants. While Li et al. 2019 showed that weak ligand binding to A subunit proteins could be detected by NMR, their genetic evidence was less convincing than the mutagenesis analyses presented in this manuscript. Furthermore, Artzi et al. provide biochemical evidence that the topology of the GerAA subunit is not consistent with germinant binding, namely that its N-terminus is cytoplasmic. The evidence they provide includes analysis GerAA accessibility to proteases and Cys-substituted GerAA subunits. However, the manuscript would be improved by expanding the discussion of these critical control experiments and clarifying the following about these experiments:

1. In the protease accessibility experiments, there is a band that appears over time at just under 25 kDa. This band suggests that at least part of GerAA is accessible to proteases. Is this a cross-reactive band? Does it appear in a Δ gerAA strain?
2. The SCAM analyses should include an NEM control to demonstrate that it is possible to prevent labeling of the indicated Cys residues by Mal-PEG if NEM is used to label free Cys regardless of the technology. For example, the authors argue that S317C is not accessible to the membrane impermeant crosslinker MTSES, but there appears to be less sample relative to the other mutants being tested and a faint band of GerAA-PEG being detected even with the MTSES treatment.

Overall, the authors data is convincing, but most of their experiments involve two biological replicates. The authors should analyze three biological replicates of the experiments shown in this manuscript to maximize the rigor of their study.

In addition, the authors could emphasize more that the mutants expand rather than switch germinant specificity, since Ala and Val are good germinants for all the GerAB germinant specificity mutants, and D-Ala still acts as a competitive inhibitor.

Some changes to the data presentation would also improve readability of the manuscript beyond expanding discussion of the SCAM experiment.

- Since the manuscript could be expanded to include more figures in the main text, I would

suggest moving Fig. S11 to the main manuscript as well as S5.

- Presenting the data in Figures S4 and S5, it would make the data easier to interpret (in my view) if the graphs show the responses of multiple mutant strains to a given amino acid condition. This would facilitate statistical testing (which was not included in the manuscript).
- Adding the side chains or space filling side chains in Figure 1 or added to Figure 3 would be helpful.
- The color difference between light blue, green, and darker green(?) is hard to distinguish in some of the graphs. Using colors with greater contrast would make the figures more readable.
- % Spore Survival for L199I appears to be missing from the table.
- Add ref 7 after the PDB structure is referred to (line 45)

Additional questions

- Did the authors test the different point mutants for their responsiveness to Leu, Ile, and Ser ligand, e.g. Fig S6 shows L199S in a Ser titration, but did the authors confirm that the L199I and V101A, etc. did not
- The V101 mutations appear to reproducibly reduce the overall optical density decrease observed during germination. The mutants may have a subtle decrease in % sporulation (Table S1). Is it possible that this mutation reduces DPA content in the cells, which is why they might have a smaller overall decrease in optical density. Alternatively, do some of these mutant spores fail to germinate when most of the population has germinated?
- How do the authors' data match with results from *B. megaterium* GerAB mutant analyses that suggested that GerAB is involved in ligand binding (Christie et al.)
- How many different mutations did the authors test (lines 109 and 123)? It would be helpful to describe in supplemental the additional residues the authors mutated.

Reviewer #2 (Remarks to the Author):

This manuscript describes bioinformatic analysis and structural modeling of the *B. subtilis* GerAB subunit of a spore germinant receptor, coupled with genetic analysis of germinant response to support the conclusion that this protein is the germinant-binding component of the receptor. The data is quite convincing and is presented clearly and convincingly.

The work is a strong contribution in the field of spore biology. While other studies have suggested a germinant-binding role for this protein, some studies have also suggested a different protein plays this role. The data presented here are more thorough and present a more definitive case than the earlier studies.

Comments:

Line 50: Some description of the ">20,000 GerAB homologs" that were analyzed to identify Evolutionary Coupled residues. This might be done in the Methods section. Were these all actual Ger receptor proteins pulled from Firmicute sequences or are some from metagenomes with species source unknown? Do these homologs range much more broadly into transporters (like GkApCT)?

Figure 1: It is partially my lack of experience in interpreting figure such as Panel 1a, but it took me quite a bit of time to figure out what this figure is showing. I am guessing that I will not be the only one. Once I worked it out, then reading the text of lines 44-54 seemed like a reasonable explanation. However, making the connections between the text and the figure was not simple. A slightly greater explanation might be nice for a large number of readers.

Figure 2b: The legend states that data from one of two biological replicates with two technical replicates is shown. Error bars are only visible for one strain. Was there no variation for other strains? Similarly in Figure 3 and some supplemental figures, there are sometimes error bars and in other cases none.

Reviewer #3 (Remarks to the Author):

Given the complex relationship between GerAA, AB and AC, it would seem appropriate for the

authors to measure the effect of mutations in GerAB on the other components of the operon. Changes in GerAB stability may affect GerAA and AC levels that subsequently impact the germination assay.

The study by Rudner et al. reports the characterisation of an important system in bacterial spore germination. Specifically, the system under investigation is composed of three proteins, GerAA, AB and AC. GerAB and AC are integral membrane proteins, whilst GerAA is extracellular. The authors use bioinformatics techniques, including routine sequence-based searches and more sophisticated amino acid covariance analysis, to identify GerAB as structurally similar to a previously characterised amino acid transporter, GkApcT. The authors create a homology model of GerAB using GkApcT as the template and proceed to validate this model by making variants of the binding site and observing how these affect the germination of the bacterial spores. The authors conclude that GerAB is a structural homologue of GkApcT and a member of the wider APC superfamily of secondary active transporters. Further, they suggest that GerAB functions as a nutrient sensor analogously to SLC38A9, another transporter cum nutrient receptor in eukaryotic cells.

Overall, the study presents an interesting hypothesis, which I believe is plausible but not wholly supported by the current data. The study has several significant weaknesses that need to be addressed in what I suspect would be considered significant revisions.

The first significant point is that the Ger system appears to be rather complex and incompletely understood. This makes deductions from the assay results complicated to understand, at least for me. The authors state that the protein levels of GerAA and AC depend on those of AB. Do the authors know how? Without a complete understanding of this link, it isn't easy to fully assess the impact of the variants in GerAB on germination. The authors should be running western blots for GerAA and AC for each of the variants they study in GerAB. For studies of this kind, it is imperative to understand the effects of mutations on all the protein components involved.

One of the major conclusions of this study is that GerAB is a specific L-Alanine sensor. However, to make this statement, the authors need to screen all the 20 amino acids for the WT protein. As it stands, the authors have provided only a small snapshot of AAs. It isn't easy to draw solid conclusions from this data without knowing how the WT and active variants respond to other AAs.

The concluding paragraph (lines 196-204, page 7) is very speculative given the preceding data. The authors suggest a model for interactions between the three subunits based on covariation analysis. For a high impact journal such as Nature Communications, I feel you should be demonstrating this using biochemical evidence, such as a Co-IP or Co-purification.

Ultimately, I struggled to see the major advancements made in this study. This could well be my lack of understanding in the field of bacterial spore germination. However, while this is an exciting system, I didn't see how this study significantly improved our knowledge above the current literature. For example, on Uniprot, GerAB is annotated as belonging to the APC superfamily, which appears to be one of the major conclusions to this study. While bioinformatic-function work is well executed, there are significant gaps in the functional data that limit the mechanistic insights into this system. The absence of any biochemical evidence that these proteins physically interact in the spore is a significant weakness and reduces the impact of the study. Does this study really move the field forward enough to warrant publication at this level? I am not convinced, but be would be happy for the authors to convince me.

Minor comments.

The authors have omitted side chains from their structural figures, which I found odd given the importance of the homology model as the basis for the functional characterisation.

Fig.1. It would be helpful to have the GkApcT structure alongside the homology model for comparison.

Fig. 2. Initially, I thought the SleB sample was the loading control for the WB. However, later in

the paper, they demonstrate that SleB is an important component in the germination pathway itself. Could the authors clarify this in the Ms?

Reviewer #4 (Remarks to the Author):

In their succinct, well-written manuscript, Artzi and coworkers use computational predictions and experiments to determine that GerAB serves as the *B. subtilis* L-alanine sensor. An initial homology model is constructed based on a solved structure of GkApcT. Cross-referencing the native contacts within this homology model and those predicted by EVcouplings suggests that GerAB and GkApcT are indeed homologs. The authors proceed through an exhaustive mutational study in which they identify point mutations that can abolish germination or change the specificity of GerAB from L-alanine to L-serine or L-leucine. Finally, the authors demonstrate that the introduction of bulky residues near the L-alanine binding pocket leads to premature germination in the absence of L-alanine.

The modeling study of GerAB appears to have been performed in line with best practice. The procedure of the authors is also largely self-checking, as their experimental studies verify the importance of many of the residues predicted by their homology model to play a role in L-alanine binding. I have no major concerns, but I would be interested to know if the authors tried any co-evolutionary contact predictors other than EVcouplings, as different methods tend to predict different types of contacts (PMID: 31693112)? Determination of which methods produce the best predictions for "functional contacts" such as those discussed in the present manuscript are of great interest.

REVIEWER COMMENTS

Reviewer #1 (Remarks to the Author):

The manuscript by Artzi et al. provides compelling evidence for a model in which the B subunit of germinant receptors serves as the direct sensor of amino acids in bacterial spores. Using evolutionary co-variation analysis, the authors discovered structural homology between GerAB subunits and the Amino acid-Polyamine-Organocation (APC) superfamily. Guided by the structure of an APC family member, GkApcT, bound to its alanine ligand, the authors target the putative ligand binding region of *Bacillus subtilis* GerAB for mutagenesis. They identify several interesting classes of mutations that impact GerAB function without affecting its levels in *B. subtilis* spores: (i) mutations that ablate germinant sensing (G25A no longer responds to alanine germinant), (ii) mutations that reduce the size of a side chain and accordingly expand germinant specificity to include specific amino acids like leucine (V101C/A), serine (L199S), or isoleucine (L199I and T287V) in addition to the primary alanine germinant, and (iii) mutations that mimic germinant binding and prematurely activate germination (V101F and T287L).

The manuscript is well written and presents intriguing possible mechanisms for germinant sensing and signal transduction informed by studies of transporters in eukaryotic organisms, so it will be of wide interest to the community. However, the title and abstract are not as circumspect in terms of acknowledging that it is theoretically possible that the mutations the authors have identified affect signal transduction rather than ligand binding. Thus, the authors should temper their language slightly.

We thank the reviewer for his/her enthusiasm for our study.

We have tempered our language in the abstract and discussion to reflect the theoretical possibility that the mutations we analyzed impact signal transduction rather than ligand binding. However, given the fact that GerAB has an APC-superfamily-like ligand binding pocket and our mutations that specifically (and narrowly) alter amino acid specificity are in residues that line this pocket, we request keeping the original title.

Despite lacking biochemical evidence of GerAB directly binding to amino acid ligands, the authors' identification of mutants with expanded germinant specificity provides strong genetic support for GerAB functioning as the germinant sensor. Their analysis of the germinant sensitivity by titrating different germinants and comparing the germinant specificity of their different mutants provides convincing data in support of hypothesis. Their work stands in contrast to previously published work (2019) implicating the A subunit of Ger receptors as the direct binding partner for germinants. While Li et al. 2019 showed that weak ligand binding to A subunit proteins could be detected by NMR, their genetic evidence was less convincing than the mutagenesis analyses presented in this manuscript. Furthermore, Artzi et al. provide biochemical evidence that the topology of the GerAA subunit is not consistent with germinant binding, namely that its N-terminus is cytoplasmic. The evidence they provide includes analysis GerAA accessibility to proteases and Cys-substituted GerAA subunits. However, the manuscript would be improved by expanding the discussion of these critical control experiments and clarifying the following about these experiments:

We appreciate the reviewer's knowledge of the dense germination literature and thank him/her for recognizing the thoroughness with which we address the discrepancies between our findings and previously published data. In the revised manuscript we have included additional control experiments and expanded our discussion of these experiments (see below).

1. In the protease accessibility experiments, there is a band that appears over time at just under 25 kDa. This band suggests that at least part of GerAA is accessible to proteases. Is this a cross-reactive band? Does it appear in a Δ gerAA strain?

Prompted by the reviewer's question, we repeated the protease accessibility experiments in our wild-type strain (WT) that lacks vegetative expression of GerAA. As shown in the reviewer figure below, the band that appears at just under 25 kDa in the experimental strain is indeed specific to GerAA and is not detected in WT cells that do not express GerAA. Thus, this band reflects some accessibility of GerAA to Proteinase K cleavage. We note that the cleavage product recognized by our antibody is significantly fainter than the full-length protein and therefore suggests that the cleavage site is only modestly accessible. Our evolutionary co-variation analysis predicts that GerAA has two short extracellular loops and we suspect that Proteinase K has limited access to one of them resulting in the limited cleavage observed. The immunoblot presented in Supplementary Figure 11 is somewhat saturated compared to the reviewer figure, giving the impression that this site is more accessible to Proteinase K cleavage. Importantly, the signal for full-length GerAA is barely diminished during the 20 min time course. In the revised manuscript we discuss this lower band in the legend to Figure S11 and thank the reviewer for requesting this control.

2. The SCAM analyses should include an NEM control to demonstrate that it is possible to prevent labeling of the indicated Cys residues by Mal-PEG if NEM is used to label free Cys regardless of the technology. For example, the authors argue that S317C is not accessible to the membrane impermeant crosslinker MTSES, but there appears to be less sample relative to the other mutants being tested and a faint band of GerAA-PEG being detected even with the MTSES treatment.

We thank the reviewer for requesting this control. We have repeated the SCAM experiments with and without NEM and have replaced the original supplemental figure (Figure S11D) with one of the three new biological replicates performed. The data clearly indicate that all of the Cys residues in GerAA tested are accessible to NEM and can prevent Mal-PEG labeling. These new data strengthen our conclusions.

Regarding S317C, our data indicate that this residue is partially accessible to the membrane impermeant crosslinker MTSES. However, this residue is predicted to be located on a small extracellular loop of GerAA that is further predicted to interact with GerAC. We think the small size of the loop and/or its interaction with GerAC accounts for its partial accessibility to MTSES. We note that the incubation time with NEM is longer (100 min) than MTSES (10 min), allowing for a more complete reaction with S317C and therefore a more complete block of Mal-PEG labeling. However and importantly, the fact that MTSES partially reacts with S317C (and partially blocks Mal-PEG labeling) while S55C and S94C do not react with this membrane impermeant reagent (and are therefore efficiently labeled with Mal-PEG) combined with our new data (requested by this reviewer) that all of the residues are blocked by NEM strongly suggests that the N-terminus of GerAA resides in the spore core.

Overall, the authors data is convincing, but most of their experiments involve two biological replicates. The authors should analyze three biological replicates of the experiments shown in this manuscript to maximize the rigor of their study.

We have now performed 3 biological replicates for all experiments in main text, 3 biological replicates for all sporulation assays, cytological analyses, immunoblots, and protease accessibility and SCAM experiments in the supplemental Figures. All supplemental germination assays have been performed with 2 or 3 biological replicates (as indicated in the legends) and 2 technical replicates per biological replicate.

In addition, the authors could emphasize more that the mutants expand rather than switch germinant specificity, since Ala and Val are good germinants for all the GerAB germinant specificity mutants, and D-Ala still acts as a competitive inhibitor.

We now emphasize this point in several places in the manuscript and do not describe the mutants as switching specificity.

Some changes to the data presentation would also improve readability of the manuscript beyond expanding discussion of the SCAM experiment.

- Since the manuscript could be expanded to include more figures in the main text, I would suggest moving Fig. S11 to the main manuscript as well as S5.

We thank the reviewer for suggesting we move Supplementary Figure 11 to the main text. While the data in Supplementary Figure 11 provide additional support for the idea that GerAB (and not GerAA) is the nutrient sensor, we have decided to keep this figure in the supplement. We think ruling out the N-terminus of GerAA as a potential nutrient sensor is important for researchers in the field and this is why we have invested significant effort in doing so. However, we think making this a central point of the paper by putting the data in the main text would be a distraction to the general reader. That being said, we have expanded our discussion of these important control experiments. We hope this is acceptable to the reviewer.

- Presenting the data in Figures S4 and S5, it would make the data easier to interpret (in my view) if the graphs show the responses of multiple mutant strains to a given amino acid condition. This would facilitate statistical testing (which was not included in the manuscript).

We appreciate the strengths of both ways of presenting the data. We compared them side-by-side and queried members of the lab about which was easier to interpret. The consensus was that the original presentation style was clearer. We therefore request to keep these figures in their current form.

- Adding the side chains or space filling side chains in Figure 1 or added to Figure 3 would be helpful.

Since the predicted GerAB structure is a threaded model, the resolution is not high enough to confidently place the side chains. We therefore felt showing them in the main figure could be misleading and/or overstate our claims. However, we have added the side chains of the relevant amino acids in GerAB and GkApcT in Supplementary Figure 1. (They look great but should be taken with a grain of salt.)

- The color difference between light blue, green, and darker green(?) is hard to distinguish in some of the graphs. Using colors with greater contrast would make the figures more readable.

While building the figures for publication, we tried several different colors for the germination assay plots. After consulting with members of our group, we felt the selected colors for these plots most clearly distinguished the different strains or conditions while retaining a modicum of aesthetics. We request keeping plots in their current coloring scheme.

- % Spore Survival for L199I appears to be missing from the table.
Thank you. Corrected.

- Add ref 7 after the PDB structure is referred to (line 45)
Added.

Additional questions

- Did the authors test the different point mutants for their responsiveness to Leu, Ile, and Ser ligand, e.g. Fig S6 shows L199S in a Ser titration, but did the authors confirm that the L199I and V101A, etc. did not

The point mutants were all tested for their responsiveness to Leu, Ile, and Ser. L199S was responsive to Ser but was unchanged in response to Leu or Ile. L199I had an increased response to Ile, but not to Ser. V101A, V101C, and L199I had increase responsiveness to Leu and Ile but were unchanged in their responsiveness to Ser. All of these assays are included in Supplementary Figure 5.

- The V101 mutations appear to reproducibly reduce the overall optical density decrease observed during germination. The mutants may have a subtle decrease in % sporulation (Table S1). Is it possible that this mutation reduces DPA content in the cells, which is why they might have a smaller overall decrease in optical density. Alternatively, do some of these mutant spores fail to germinate when most of the population has germinated?

The reviewer is correct. The overall optical density decrease in our germination assays with V101A and V101C is reproducibly less than WT. We are not certain of the cause, but we think these mutants are more prone to spontaneous germination. Thus, during or after purification of the phase-bright spores we think a subset spontaneously germinate resulting in a more modest decrease in optical density in our assay. This is, in part, why we analyzed and presented phase-contrast images of these spores during germination. We have added this point in the Figure Legend which now reads:

"the overall decrease in OD in the V101 mutants is reproducibly less than WT, likely due to spontaneous germination of a subset of the spores during purification for these assays."

- How do the authors' data match with results from *B. megaterium* GerAB mutant analyses that suggested that GerAB is involved in ligand binding (Christie et al.)

We thank the reviewer for asking. We have tried to compare our mutants to the studies on GerVB from the Christie lab. Unfortunately, their analysis was confined to the known germinants of this receptor. Furthermore GerVB (with its cognate subunits GerUA and GerUC) responds to 4 germinants, including two amino acids (Glucose, Proline, Leucine, K+) making it difficult assign specific residues to individual germinants. We are definitely interested in following up on this type of analysis in the future.

- How many different mutations did the authors test (lines 109 and 123)? It would be helpful to describe in supplemental the additional residues the authors mutated.

We thank the reviewer for his/her interest in the full set of mutants tested. We now include a list of mutants that we tested but did not include in our detailed analysis in Supplemental Table 2. We also included a brief description of their phenotypes in regard to spore viability and germination response.

Reviewer #2 (Remarks to the Author):

This manuscript describes bioinformatic analysis and structural modeling of the *B. subtilis* GerAB subunit of a spore germinant receptor, coupled with genetic analysis of germinant response to support the conclusion that this protein is the germinant-binding component of the receptor. The data is quite convincing and is presented clearly and convincingly.

The work is a strong contribution in the field of spore biology. While other studies have suggested a germinant-binding role for this protein, some studies have also suggested a different protein plays this role. The data presented here are more thorough and present a more definitive case than the earlier studies.

We thank the reviewer for his/her enthusiasm for our work.

Comments:

Line 50: Some description of the ">20,000 GerAB homologs" that were analyzed to identify Evolutionary Coupled residues. This might be done in the Methods section. Were these all actual Ger receptor proteins pulled from Firmicute sequences or are some from metagenomes with species source unknown? Do these homologs range much more broadly into transporters (like GkApcT)?

We thank the reviewer for requesting this information. As described below, the overwhelming majority of Ger receptors proteins were from Firmicutes. Our analysis suggests that the homologs used do not overlap with transporters like GkApcT. We have added the information below to the Supplementary Methods section.

For the GerAB monomer, we used a 2019 Uniref100 database to build the alignment using a 5-iteration jackhmmer protocol with domain and sequence bitscore thresholds both set to 0.3*(the length of the protein). The resulting alignment primarily contains protein sequences with 'spore germination' in the description, but there are some proteins of unknown function and other annotations. Monomer evcouplings analysis was performed directly on the resulting alignment after removing sequences with more than 50% gaps compared to the *B. subtilis* GerAB sequence. Additionally, for the GerAB-GerAA complex interaction predictions only, we filtered the monomer alignments for these two proteins so that we were only considering sequences from species with GerAA and GerAB hits within 10,000 nucleotides of each other on the genome.

For species included in the GerAB monomer alignment, the overwhelming majority (19,986 out of 20,525 sequences) are from Firmicutes. A small fraction from eukaryotes and other non-Firmicutes bacteria were recovered. The alignment did not have significant overlap with the GkApcT family based on domain analysis. The sequence for PDB 5oqt, a GkApcT transporter with Uniprot ID Q5L1G5 that was used to build the homology model for GerAB, was not present in the alignment used for the evcouplings analysis. Furthermore, the alignment sequences were mapped to 12,016 SwissProt + TrEMBL IDs, which was then cross-referenced against the Pfam domain database to assess overlap between amino acid transporter domains and the alignment. All of the alignment sequences either have no domain annotated or have Pfam ID PF03845 (spore germination protein). No proteins annotated with Pfam ID PF13520 (amino acid permease), the domain associated with the GkApcT structure were detected.

Figure 1: It is partially my lack of experience in interpreting figure such as Panel 1a, but it took me quite a bit of time to figure out what this figure is showing. I am guessing that I will not be the only one. Once I worked it out, then reading the text of lines 44-54 seemed like a reasonable explanation. However, making the connections between the text and the figure was not simple. A slightly greater explanation might be nice for a large number of readers.

We thank the reviewer for pointing this out. We have expanded our explanation of Panel 1a and included a more detailed explanation in the legend to Supplementary Figures 1.

Figure 2b: The legend states that data from one of two biological replicates with two technical replicates is shown. Error bars are only visible for one strain. Was there no variation for other strains? Similarly in Figure 3 and some supplemental figures, there are sometimes error bars and in other cases none.

All graphs have error bars. In places where error bars are not detectable, it is because the symbols are larger than the error bars.

Reviewer #3 (Remarks to the Author):

Given the complex relationship between GerAA, AB and AC, it would seem appropriate for the authors to measure the effect of mutations in GerAB on the other components of the operon. Changes in GerAB stability may affect GerAA and AC levels that subsequently impact the germination assay.

Prompted by the reviewer's concern, we have performed immunoblots on spores for all the GerAB mutants discussed in the manuscript. These immunoblots are presented in Supplementary Figure 7. As anticipated, most of the mutants that were germination proficient had similar levels of GerAA and GerAC-His compared to WT. The two mutants (G200A and Y291S) that phenocopied the *gerAB* null mutant had very low levels of GerAA and undetectable levels of GerAC-His - similar to a strain lacking GerAB. These are discussed in the main text. The V101A and V101C mutants that respond to L-Leucine had somewhat reduced levels of GerAA and GerAC-His but were well above the levels in a strain lacking GerAB. The V101F mutant that constitutively germinates had even lower levels of GerAA and GerAC-His compared to V101A and V101C. This mutant triggers premature germination during sporulation and we are analyzing GerAA and GerAC levels in dormant spores (in a Δ sleB mutant background to prevent degradation of the spore cell wall). It is therefore possible that the levels of these proteins were higher during sporulation. Alternatively, the low level of the germination complex observed is sufficient to trigger premature germination. Importantly, the other mutant that triggered premature germination (T287L) had levels of GerAA and GerAC that were similar to WT. Importantly, none the GerAB mutants resulted in higher levels of GerAA and GerAC. Thus, the expanded nutrient specificity caused by mutations in the putative ligand-binding pocket are not due to increased levels of GerA receptor complexes leading to loss of specificity. Instead, the data are most easily explained by alterations in nutrient recognition.

The study by Rudner et al. reports the characterisation of an important system in bacterial spore germination. Specifically, the system under investigation is composed of three proteins, GerAA, AB and AC. GerAB and AC are integral membrane proteins, whilst GerAA is extracellular. The authors use bioinformatics techniques, including routine sequence-based searchers and more sophisticated amino acid covariance analysis, to identify GerAB as structurally similar to a previously characterised amino acid transporter, GkApcT. The authors create a homology model of GerAB using GkApcT as the template and proceed to validate this model by making variants of the binding site and observing how these affect the germination of the bacterial spores. The authors conclude that GerAB is a structural homologue of GkApcT and a member of the wider APC superfamily of secondary active transporters. Further, they suggest that GerAB functions as a nutrient sensor analogously to SLC38A9, another transporter cum nutrient receptor in eukaryotic cells.

Overall, the study presents an interesting hypothesis, which I believe is plausible but not wholly supported by the current data. The study has several significant weaknesses that need to be addressed in what I suspect would be considered significant revisions.

The first significant point is that the Ger system appears to be rather complex and incompletely understood. This makes deductions from the assay results complicated to understand, at least for me. The authors state that the protein levels of GerAA and AC depend on those of AB. Do the authors know how? Without a complete understanding of this link, it isn't easy to fully assess the impact of the variants in GerAB on germination. The authors should be running western blots for GerAA and AC for each of the variants they study in GerAB. For studies of this kind, it is imperative to understand the effects of mutations on all the protein components involved.

We now provide immunoblots for GerAA and GerAC for all GerAB mutants tested. These can be found in Supplementary Figure 7 and are described above. We thank the reviewer for requesting this analysis. We think it strengthens our conclusions and the impact of the study.

In regards to the link between the three GerA proteins, it has not been established how GerAA and GerAC depend on GerAB for stability. The simplest explanation for this co-dependency is that the three proteins interact and in the absence of any one of the three, the remaining two components become susceptible to proteolysis. The evolutionary co-variation analysis presented in Supplementary Figure 11 is consistent with their interaction but does not prove it. Further support comes from the immunoblots requested by this reviewer (Supplementary Figure 7) in which GerAB mutants that result in reduced levels of GerAA have corresponding reductions in GerAC. As discussed above, all the mutants with expanded nutrient specificity had levels of GerAA and GerAC that were comparable to or lower than WT. Accordingly, these findings argue that the expanded specificity is not due to increased levels of nutrient receptors and instead suggest the GerAB mutations in the putative ligand-binding pocket are likely to be affecting nutrient recognition.

One of the major conclusions of this study is that GerAB is a specific L-Alanine sensor. However, to make this statement, the authors need to screen all the 20 amino acids for the WT protein. As it stands, the authors have provided only a small snapshot of AAs. It isn't easy to draw solid conclusions from this data without knowing how the WT and active variants respond to other AAs.

We thank the reviewer for requesting clarification. We were not clear enough in the original submission. Spores harboring the *gerA* locus can respond to micromolar concentrations of L-Ala and low millimolar concentrations of L-Val. A subset of additional amino acids (e.g. L-Isoleucine, L-Cysteine, and L-Glutamate) can induce germination in a GerA-dependent manner but require concentrations in the 30-50 mM range. All 20 amino acids were tested in the late 60's and early 70's (Wax and Freese 1968; and Nitta et al, 1974). When we initiated this study we screened 19 amino acids with WT spores (see reviewer figure below). Based on our findings, we decided to focus on a subset of amino acids for our study. We now mention this point more explicitly in the text. It now reads:

"Spores harboring the wild-type *gerA* locus respond to micromolar concentrations of L-alanine, low millimolar concentrations of L-valine and inefficiently to a few other amino acids when present at concentrations >25mM^{26,27}. Here, we tested a panel of 9 amino acids."

The concluding paragraph (lines 196-204, page 7) is very speculative given the preceding data. The authors suggest a model for interactions between the three subunits based on covariation analysis. For a high impact journal such as Nature Communications, I feel you should be demonstrating this using biochemical evidence, such as a Co-IP or Co-purification.

The concluding paragraph focuses on the possible function of GerAC in the complex. As part of our argument we present evolutionary co-variation data for the three subunits. We do not think our discussion is overly speculative nor do we think providing biochemical evidence for a complex is required to argue that GerAB is likely to be the nutrient sensor and/or that GerAC could function to channel L-alanine to the ligand-binding pocket. However, it is definitely our long-term goal to purify and characterize this putative germinant receptor complex.

Ultimately, I struggled to see the major advancements made in this study. This could well be my lack of understanding in the field of bacterial spore germination. However, while this is an exciting system, I didn't see how this study significantly improved our knowledge above the current literature. For example, on Uniprot, GerAB is annotated as belonging to the APC superfamily, which appears to be one of the major conclusions to this study. While bioinformatic-function work is well executed, there are significant gaps in the functional data that limit the mechanistic insights into this system. The absence of any biochemical evidence that these proteins physically interact in the spore is a significant weakness and reduces the impact of the study. Does this study really move the field forward enough to warrant publication at this level? I am not convinced, but we would be happy for the authors to convince me.

Spore germination has been studied for over a century. The discovery that *ger* loci (like *gerA*) are required for spores to respond to specific nutrients was first reported almost 40 years ago. Since that time there have been >100 papers on these loci with little progress in establishing whether or not they are indeed nutrient receptors and, if so, which subunit is the sensor. As described in the Introduction, there have been papers arguing that the A subunit is the sensor while others have argued that the B subunit serves this role. The data in these studies to support either claim were indirect in most instances limited. Our identification of point mutations in residues that line the ligand binding pocket found in other APC family members that narrowly expand nutrient specificity or inappropriately trigger germination without stimulus represents the strongest and most compelling evidence that the *ger* loci are indeed germinant receptors and that GerAB is the nutrient sensor. We firmly believe that this paper moves the field forward and warrants publication in Nature Communication.

Minor comments.

The authors have omitted side chains from their structural figures, which I found odd given the importance of the homology model as the basis for the functional characterisation.

Since the predicted GerAB structure is a threaded model, the resolution is not high enough to confidently place the side chains. We therefore felt showing them in the main figure could be misleading and/or overstate our claims. However, we have added the side chains of the relevant amino acids in GerAB and GkApcT in Supplementary Figure 1.

Fig.1. It would be helpful to have the GkApcT structure alongside the homology model for comparison.

We include the GkApcT structure alongside the homology model in Supplementary Fig. 1. We have also added the relevant sidechains to both structures.

Fig. 2. Initially, I thought the SleB sample was the loading control for the WB. However, later in the paper, they demonstrate that SleB is an important component in the germination pathway itself. Could the authors

clarify this in the Ms?

The reviewer is correct. SleB is a cell wall hydrolase that is synthesized in the developing spore where it accumulates in an inactive state. Upon nutrient detection, one of the downstream steps in the germination pathway is the activation of SleB. Degradation of the specialized spore peptidoglycan allows the spore to initiate vegetative growth. Although SleB is in the germination pathway its production and stability in dormant spores are unaffected by the presence or absence of the Ger receptors. Accordingly, it is a commonly used loading control. We have clarified this point in the methods section. It now reads:

"Although SleB is in the germination pathway its production and stability in dormant spores are unaffected by the presence or absence of the Ger receptors. Accordingly, it serves as a reliable control for loading."

Reviewer #4 (Remarks to the Author):

In their succinct, well-written manuscript, Artzi and coworkers use computational predictions and experiments to determine that GerAB serves as the *B. subtilis* L-alanine sensor. An initial homology model is constructed based on a solved structure of GkApcT. Cross-referencing the native contacts within this homology model and those predicted by EVcouplings suggests that GerAB and GkApcT are indeed homologs. The authors proceed through an exhaustive mutational study in which they identify point mutations that can abolish germination or change the specificity of GerAB from L-alanine to L-serine or L-leucine. Finally, the authors demonstrate that the introduction of bulky residues near the L-alanine binding pocket leads to premature germination in the absence of L-alanine.

We thank the reviewer for his/her enthusiasm for our work and for the quality of the presentation.

The modeling study of GerAB appears to have been performed in line with best practice. The procedure of the authors is also largely self-checking, as their experimental studies verify the importance of many of the residues predicted by their homology model to play a role in L-alanine binding. I have no major concerns, but I would be interested to know if the authors tried any co-evolutionary contact predictors other than EVcouplings, as different methods tend to predict different types of contacts (PMID: 31693112)? Determination of which methods produce the best predictions for "functional contacts" such as those discussed in the present manuscript are of great interest.

Thank you for this question - we agree that predicting which contacts are functional under different conditions is of great interest, beyond just predicting structure alone. Here, we only used the direct couplings analysis (DCA) EVcouplings to make these predictions, and did not test other tools that are more focused on fold prediction than on identifying the most functional residue interactions.

REVIEWERS' COMMENTS

Reviewer #1 (Remarks to the Author):

The revised manuscript addresses my concerns and the concerns raised by other Reviewers (in my opinion). The added control experiments and additional replicates increase the rigor of the study, and the slight wording changes to the manuscript further improve its readability. Artzi et al's work is beautiful, and the manuscript is well written and compelling. The manuscript greatly advances our understanding of how germinants are sensed by many bacterial spores and identifies a number of exciting avenues to pursue in future analyses.

Reviewer #3 (Remarks to the Author):

The authors have carried out several additional experiments that support their previous results. These have been nicely folded into the previous Ms, along with additional text and contextual changes that substantially strengthen the study. The additional WB data in Supplementary Figure 7 supports their hypothesis on the co-dependency of Ger AA and Ger AC on the stability/presence of Ger AB - a major hypothesis of the study. I, therefore, support publication in the present form.